# Validity of Three-Dimensional Tortuous Pore Structure and Fouling of Hemoconcentration Capillary Membrane Using the Tortuous Pore Diffusion Model and Scanning Probe Microscopy

**DOI:** 10.3390/membranes10110315

**Published:** 2020-10-29

**Authors:** Makoto Fukuda, Hiroki Yoshimoto, Hitoshi Saomoto, Kiyotaka Sakai

**Affiliations:** 1Biomedical Engineering, Kindai University, 930 Nishimitani, Kinokawa-City, Wakayama 649-6493, Japan; 1618360001w@waka.kindai.ac.jp; 2Industrial Technology Center of Wakayama Prefecture, 60 Ogura, Wakayama-City, Wakayama 649-6261, Japan; saomoto@wakayama-kg.jp; 3Chemical Engineering, Waseda University, 3-4-1 Okubo, Shinjuku-ku, Tokyo 169-8555, Japan; kisakai@waseda.jp

**Keywords:** tortuous pore diffusion model (TPD model), scanning probe microscopy (SPM), capillary, hollow fiber membrane, three-dimensional tortuous pore

## Abstract

Hemoconcentration membranes used in cardiopulmonary bypass require a pore structure design with high pure water permeability, which does not allow excessive protein adsorption and useful protein loss. However, studies on hemoconcentration membranes have not been conducted yet. The purpose of this study was to analyze three-dimensional pore structures and protein fouling before and after blood contact with capillary membranes using the tortuous pore diffusion model and a scanning probe microscope system. We examined two commercially available capillary membranes of similar polymer composition that are successfully used in hemoconcentration clinically. Assuming the conditions of actual use in cardiopulmonary bypass, bovine blood was perfused inside the lumens of these membranes. Pure water permeability before and after bovine blood perfusion was measured using dead-end filtration. The scanning probe microscopy system was used for analysis. High-resolution three-dimensional pore structures on the inner surface of the membranes were observed before blood contact. On the other hand, many pore structures after blood contact could not be observed due to protein fouling. The pore diameters calculated by the tortuous pore diffusion model and scanning probe microscopy were mostly similar and could be validated reciprocally. Achievable pure water permeabilities showed no difference, despite protein fouling on the pore inlets (membrane surface). In addition, low values of albumin sieving coefficient are attributable to protein fouling that occurs on the membrane surface. Therefore, it is essential to design the membrane structure that provides the appropriate control of fouling. The characteristics of the hemoconcentration membranes examined in this study are suitable for clinical use.

## 1. Introduction

Hemoconcentration membranes used in extracorporeal circulatory surgeries require a membrane pore structure design with a high pure water permeability, which does not allow useful proteins such as albumin to leak out [1,2]. The most serious dysfunction of a hemoconcentrator is abnormally decreased pure water permeability due to protein fouling on a membrane surface. If critical protein fouling occurs, pure water permeability decreases, so the circuit including a hemoconcentrator must be replaced during the extracorporeal cardiopulmonary bypass (CPB) surgery. During this replacement, the CPB surgery is interrupted. This is a serious accident that puts the patient’s life in danger. However, studies on hemoconcentration membranes have not been conducted yet, and therefore we are interested in pore structures before and after blood contact due to protein fouling [3].

Semipermeable membranes (hollow fiber membrane or flat sheet membrane) have porous structures, and three-dimensional (3D) laminated structures of polymer particles (Figure 1) refer to the model, according to publication [4]. The membrane is composed of polymer particles, and there are three types of pores (independent pore, through pore, semi-through pore). The solute permeates inside the through pore. The membrane including through pores and semi-through pores is a three-dimensional (unevenness) structure, and through pores and semi-through pores are tortuous, making the permeate distance greater than the thickness of the membrane [5,6,7,8]. Therefore, it is important to observe and analyze the 3D pore structure opened in the membrane and measure the mass transfer rate (permeability) of the solute, and to analyze the relation between 3D pore structure and transport phenomena in a membrane. The tortuous pore diffusion model (TPD model) is significant in the analysis of transport phenomena in a membrane [5,6,7,8]. In the TPD model, the higher-order structure factors of the membrane are associated with solute permeability, making it useful for estimating the permeability of any substances, analyzing the mass transfer phenomenon in a membrane, and designing a new blood purification membrane. 

Observation and analysis methods of the pore structures of membranes can be broadly divided into indirect methods such as evapoporometry, nanoperm porometry, and nitrogen adsorption and direct methods such as scanning electron microscopy (SEM), field emission SEM (FESEM), environmental SEM (ESEM), transmission EM (TEM), and atomic force microscopy (AFM).

The evapoporometric (EP) method, which is an indirect method, is used to analyze pore size distribution of ultrafiltration membranes for water treatment membranes [9,10]. EP can be easily applied in symmetric membranes; however, in the case of asymmetric membranes having skin and supporting layers, the pore diameter of the skin layer (inner surface pores of inner lumen of the capillary membranes) needs to be determined by another approach and the model for predicting pore shape. Furthermore, an operation of the EP device is very complicated, and it takes about 8 h to measure one sample.

SEM and FESEM require a third material coating on the sample surface. Since FESEM uses a gas ionization detector, surface coating is not required. Although FESEM has been applied to pore size distribution measurement of ultrafiltration membranes, the resolution must be further increased [9,10]. In contrast, AFM allows direct observation with very high resolution and without any special sample preparation [9,10,11]. However, only a very small area of less than 1 µm^2^ can be observed; thus, a problem exists when applying this method to membranes (such as those made by the phase separation method) that do not have well-defined pores. Furthermore, these devices are not readily available or versatile for these direct observations [9,10].

Most medical membranes for various blood purifications are asymmetric membranes. Moreover, the properties of the innermost surface of inner lumen of the capillary membrane, in particular, have a notable effect on solute permeability and biocompatibility of the membrane, showing that the pore structure of the innermost surface of the membrane is closely related to solute permeability [11,12,13,14]. Therefore, when identifying a blood purification membrane for medical use, clarifying the innermost surface pore structure of the membrane is of significance. Thus, the direct methods, such as using AFM, are more suitable for observing most medical membranes compared to the indirect methods.

Hayama et al. attempted to successfully observe nano-order pore structures directly on hemodialysis membranes using AFM [15,16,17]. Hayama et al. were the first to clarify the AFM image of surface pores of a dialysis membrane, but it was insufficient to analyze 3D concave structures. Therefore, we focused on dynamic force microscopy (DFM), which is more appropriate for observing soft samples and samples with large irregularities compared to conventional AFM. We could successfully observe the 3D tortuous pore structures of hemoconcentration membranes and measure pore diameter and distribution [3,18].

In this study, we analyze 3D pore structures and fouling of capillary hemoconcentration membranes using the TPD model and a scanning probe microscopy (SPM) system (DFM). The targets are commercially available hemoconcentration membranes that are being produced successfully in the industry and used clinically. We discuss the validity of pore diameters calculated by the TPD model and SPM, and relations between changes of 3D pore structure and typical mass transfer properties due to protein fouling.

## 2. Materials and Methods

### 2.1. Hemoconcentration Membrane (Hemoconcentrator)

The samples studied were commercially available hollow fiber hemoconcentrator membranes PUREMA A and B (JMS Co., Ltd., Hiroshima, Japan; 3M Co. Ltd., Minneapolis, MN, USA), with a asymmetric structure. These are capillary membranes of similar polymer composition (polyethersulfone), which are successfully used clinically for hemoconcentration. The membranes, especially inner surfaces of the membranes, are hydrophilized by blending poly(vinylpyrrolidone) (PVP), a hydrophilizing agent, to prevent surface fouling or pore blocking due to protein adsorption, and obtain biocompatibility. Table 1 shows the specification of the hollow fiber hemoconcentrator membranes. The pure water permeabilities of these membranes were significantly different, and the membranes were expected to be different in pore structure. Here, isolated single fibers were investigated instead of those in their housing.

### 2.2. Bovine Blood Perfusion Experiment, Measurement of Sieving Coefficient (SC)

We examined the hemoconcentration membranes PUREMA^®^ A and B (3M Co. Ltd., hemoconcentrator: Aquastream^®^ JMS Co., Ltd., Hiroshima, Japan). Assuming the conditions of actual use in extracorporeal circulation and diluted human blood property [2], bovine blood (Hct = 23%, Hb = 7.5 g/dL, 37 °C) containing heparin Na was perfused in hemoconcentrator devices (membrane area: 1.1 m^2^) for 10 min under 2 conditions (*n =* 10 each) (high load: flow rate of blood side *Q*_B_ of 500 mL/min, filtration flow rate *Q*_F_ of 100 mL/min, and transmembrane pressure (TMP) of approximately 150 mmHg; low load: flow rate of blood side *Q*_B_ of 100 mL/min, filtration flow rate *Q*_F_ of 30 mL/min, and transmembrane pressure (TMP) of approximately 75 mmHg). 

The pure water permeability before and after bovine blood perfusion was measured by the STOP method (dead-end filtration) using sterile reverse osmosis (RO) water [19,20] defined by the Japanese Society for Artificial Organs performance evaluation criteria. The temperature of the RO water within a thermostatic chamber was 310 K. We irrigated RO water at about TMP = 50, 75, 100 mmHg, Q_F_ = 150, 250, 330 mL/min (1.1 m^2^, 1 min), and pure water permeability was calculated. The pure water permeability of the sample after blood contact was measured, immediately after bovine blood circulation. Data on the hemoconcentrator membranes were treated using a paired t-test (*p* < 0.05).

After bovine blood perfusion, 2% glutaraldehyde aqueous solution using sterile RO water was perfused (200 mL/min, 5 min) into the module and drained, and subsequently allowed to stand for approximately 60 min for protein immobilization to take place, and then used to observe the pore structure. It must be considered that 2% glutaraldehyde aqueous solution was applied for both protein fixation and immobilization on the membrane inner surface. Glutaraldehyde has an effect on protein immobilization and an effect on protein adsorption to a membrane, and it is necessary to separate and discuss these effects. Sterile RO water was used for the rinsing process because bacterial contamination might impact either result.

The sieving coefficient (SC) of bovine serum albumin was measured using the method reported in our previous study (*n =* 3) [3]. Then, 1 h after starting the irrigation, the blood side inlet concentration, outlet concentration, and filtrate side concentration of the solute were measured.

### 2.3. Observation of Three-Dimensional Tortuous Pore Using a SPM System

The method reported in our previous study [3,18] was employed to observe the inner surface of hollow fiber membranes. Before bovine blood contact, the hemoconcentrators were washed by feeding sterile RO water at 200 mL/min for 30 min. Hollow fiber membranes were extracted from the hemoconcentrators. One fiber was cut into a length of 1 mm and approximately about 1/16th in the longitudinal direction using a razor. The samples for the purpose of observing the inner surface were cut so that their cylindrical shape was maintained and not strained during observation. However, this sample preparation is very difficult, and sample preparation yield and observation yield are still low. The creation of this observation sample is also a key point in this study. After that, three-dimensional tortuous pores were observed on the surface of inner lumen of the capillary membranes. 

### 2.4. Observation of Three-Dimensional Tortuous Pores Using Field Emission Scanning Electron Microscope (FE-SEM) 

For a comparative verification against the SPM, followed by the design validation, we used a field emission scanning electron microscope (FE-SEM) (JSM-7610F, Jeol Ltd., Tokyo, Japan) to observe the inner surfaces of the hollow fiber membranes at an accelerating voltage of 1.5 kV and an emission current of 47.2 μA. Table 2 shows the observation conditions of FE-SEM. To accurately observe the hollow fiber intramembrane surface, no conductive treatment with Au or C was applied.

## 3. Results

### 3.1. Observation of Three-Dimensional Tortuous Pores of Inner Lumen Surface of the Capillary Membranes

Figure 2(a1–a3) shows the results of the observations of PUREMA A (wet) before blood contact. Figure 2(b1–d3) shows the results of the observations of PUREMA A (wet) after blood contact, low load in blood contact (b, c) and high load in blood contact (d). Figure 3(a1–a3) shows the results of the observations of PUREMA B (wet) before blood contact. Figure 3(b1–c3) shows the results of the observations of PUREMA B (wet) after blood contact, low load in blood contact (b, c) and high load in blood contact (d). In addition, (1), (2) and (3) show the scanning areas of 2000 nm × 2000 nm, 500 nm × 500 nm, and 200 nm × 200 nm, respectively. The image of (3) shows magnified images of the pores enclosed by the blue square in (2), and the image of (2) shows magnified images of the pores enclosed by the blue square in (1). The color gradient bar at the bottom represents the scale in the Z-direction of the image. The probe tip height was 12.5 μm, and thus the height difference of the inner surface could be measured if it was under 12.5 μm.

Clear 3D pore structures and fiber structures made of polymer particles were observed before blood contact, and the pore diameter could be measured from this using the SPM line analysis (Figure 2(a1–a3) and Figure 3(a1–a3)). On the contrary, regarding the ultrafiltration membrane with high pure water permeability (L_p_) after blood contact, the pores seemed to be covered by a protein adsorption layer; hence, clear pores could not be observed (Figure 2(b1–d3)). Some of the pores were not completely covered (Figure 2(c1–c3)). After blood contact, the high load protein adsorption layer appeared to be thicker than the low load layers (Figure 2(b1–c3) vs. Figure 2(d1–d3)). 

On the other hand, regarding the ultrafiltration membrane with low L_p_ (pure water permeability), some pores were not covered by a protein adsorption layer (Figure 3(b1–c3)). The protein adsorption layer of the high and low loads of the ultrafiltration membrane with low L_p_ did not seem to differ in morphology (Figure 3(b1–b3) vs. Figure 3(c1–c3)). Furthermore, the protein adsorption layer of the ultrafiltration membrane with low L_p_ appeared to be thinner than that of the ultrafiltration membrane with high L_p_ (Figure 2(b1–d3) vs. Figure 3(b1–c3)). Under the same conditions of transmembrane pressure and flow rate of blood side, the load on the ultrafiltration membrane with low L_p_ was larger than that on ultrafiltration membrane with high L_p_, so it has been assumed that the thickness of the protein adsorption layer of the ultrafiltration membrane with low L_p_ was thicker, but the result was the opposite. The degree of hydrophilicity of low L_p_ was higher due to some differences in properties, so the amount of protein adsorption would be lower [21]. In order to accurately measure the thickness of the protein adsorption layer, it is necessary to perform a cross-sectional analysis in detail.

Several previous studies on anti-fouling and biocompatibility have elucidated the platelet activation process or leukocyte factor activation using SEM photographs [22,23,24]. However, they have not shown protein adsorption layers actually fouling innermost surface pores of inner lumen of the capillary membrane in the order of nanometers. In this order, the contact of materials with blood leads to the following reactions: (1) protein adsorption, (2) denaturation of adsorbed protein, (3) cell adhesion and extension, (4) intracellular reaction, and (5) intercellular reaction. The materials control (1)–(3), which accounts for the first half of the reactions. Important physicochemical factors include long-range interactions such as hydrophobic and electrostatic interactions. In the latter half of the reactions, namely (4) and (5), the short-range interaction represented by the key and keyhole model is dominant [25]. In the fouling process, plasma and proteins are first infiltrated and adsorbed on the surface of inner lumen of the capillary membranes. As shown in Figure 2 and Figure 3, this is the first time that the protein adsorption layer has been confirmed to foul most of the surface pores of inner lumen of the capillary membrane; hence, the finding is highly novel. For a more accurate mechanism, it is necessary to quantify the amount of protein adsorbed in the membrane and analyze membranes by obtaining temporal profiles of blood contact using method such as QCM-D. The QCM-D device tracks the periodic changes and dissipation of molecules when they bind and interact. It is on the QCM-D sensor that vibrates periodically. Therefore, using the QCM-D method, it may be possible to analyze changes in the amount of adsorbed protein per unit surface area (ng/cm^2^) and the thickness of the protein adsorption layer (µm) over time.

### 3.2. Determination of Three-Dimensional Tortuous Pore Diameter and Distribution Using a Line Analysis

For a pore diameter measurement, pores that can be measured in observation fields of 500 nm × 500 nm and 200 nm × 200 nm in size were analyzed. As none of the pores were true circles, the major and minor axis of the pores were measured through a line length analysis, and the equivalent area was calculated. Figure 4 shows the distributions of pore diameters.

Compared with the ultrafiltration membrane with low L_p_, the ultrafiltration membrane with high L_p_ membrane tended to be larger in pore diameter and pore area (Figure 4(a1,b1,a3,b3)). The pore diameter and area distributions were also shown to appear large and wide (the pore diameter and area distributions were sharp). When two membranes were compared, we found that the pore structures were different. The polyethersulfone membranes are manufactured to suit each individual use and are superior in terms of membrane productivity and versatility. 

In order to evaluate the pore structures in detail, the present analysis using this software requires a high number of samples, which means more accurate pore size distribution can be obtained. Therefore, measurement accuracy must be improved by increasing the number of data.

### 3.3. FE-SEM Observations of the Inner Surfaces of the Membrane

Figure 5 shows the results of observations of the inner surfaces of (a) PUREMA A and (b) PUREMA B. Here, (a) and (b) show a scanning area of 1200 nm × 900 nm. Observations were conducted in the above measurement areas for comparison, similar to those shown in Figure 2. Figure 5(a2,b2,a3,b3) shows the results of the observations after blood contact, low load in blood contact (a2, b2) and high load in blood contact (a3, b3). 

The differences in tortuous pore structures before and after blood contact were similar to the results in Figure 2. After blood contact, the high load protein adsorption layers were thicker than the low load layers (Figure 5(a2) vs. Figure 5 (a3), Figure 5(b2) vs. Figure 5 (b3)). Furthermore, the protein adsorption layer of the ultrafiltration membrane with low L_p_ (pure water permeability) was thinner than that of the ultrafiltration membrane with high L_p_ (Figure 5(a3) vs. Figure 5(b3)). On the other hand, regarding the ultrafiltration membrane with low L_p_, many pores were covered by a protein adsorption layer (Figure 5(b3)). Unlike the features of the SPM images, the protein adsorption layers of the high and low loads of the ultrafiltration membrane with low L_p_ seem to differ in morphology (Figure 5(b2,b3)). 

However, when we measured the magnification at 1:30,000–100,000 at an accelerating voltage of 1.5 kV, surface irregularities were confirmed to change during the measurement. It was thought that the samples were warmed by the electron beam, and that this tendency was more pronounced at a higher magnification. Consequently, we thought that, as shown in Figure 5, pores smaller than almost 100 nm had contracted, and fiber-like pores in the longitudinal direction were observed. In addition, from the FE-SEM results here, SPM is thought to be more useful than FE-SEM in measuring pore diameters of less than 30 nm, because small pores are buried and become invisible, whereas larger pores are reduced in size. Each characterization method has benefits and limitations [26].

## 4. Discussion

### 4.1. Change in Pore Structure and Mass Transfer Property Due to Protein Fouling 

Table 3 shows the measured values of the pure water permeability before and after blood contact and the sieving coefficient (SC) of albumin. As shown in Section 3.1, most surface pores of the inner lumen of the capillary membranes (PUREMA A) after blood contact were covered with the protein adsorption layer. There were highly novel differences in tortuous pore morphologies between unfouled and fouled surfaces of inner lumen of the capillary membranes (Figure 6). However, Table 3 shows that the value of the pure water permeability after blood contact was slightly lower compared to the value before blood contact. Therefore, achievable pure water permeabilities showed no difference, despite protein fouling on the pore inlets (membrane surface). On the other hand, appropriate protein fouling enables low values of albumin SC. These mass transfer properties of the hemoconcentration membrane (high pure water permeability and low albumin SC) are suitable for clinical use.

Especially for the ultrafiltration membrane with low L_p_, the pure water permeability of low load after blood contact was even slightly higher than that before blood contact, but the difference was not significant. This is because the protein adsorption layer of the high and low loads of the ultrafiltration membrane with low L_p_ (pure water permeability) were thinner than that of the ultrafiltration membrane with high L_p_ (Figure 2(b1–d3) vs. Figure 3(b1–c3)), and some pores were not covered by a protein adsorption layer (Figure 3(b1–c3)). 

Although there was no significant difference between the values of the albumin SC of the ultrafiltration membrane with low L_p_ and high L_p_, the values of low L_p_ were slightly higher compared to those of high L_p_. This might be also because some pores were not covered by a protein adsorption layer (Figure 3(b1–c3)). 

### 4.2. Tortuous Pore Diffusion Model (TPD Model)

The first membrane permeation model based on a porous structure model is the pore model proposed by Pappenheimer et al. [27]. They proposed the pore theory for the pore model to quantitatively discuss the solute permeability of kidney glomerular filtration membranes (biological membranes) (1951). Verniory et al. compared Pappenheimer’s pore model with Kedem and Katchalsky’s non-equilibrium thermodynamic friction model (1958, 1961 [28,29]) and modified Pappenheimer’s pore model to create a new pore model (1973) [30]. Subsequently, Klein et al. evaluated the solute permeability of commercially available hollow fiber dialysis membranes and attempted to analyze the membrane pore structure based on Verniory’s pore model. They created the equations of the relation between the pore radius and membrane permeability [31,32]. Klein has also contributed to the research achievements in this field with prominent researchers such as Liao and Hardy (2005 [33]).

Sakai et al. proposed the tortuous pore diffusion model by introducing tortuosity (*τ*) into the Verniory’s pore model [5,6]. The length (*L* = *τ*Δ*x*) through which the solute and water actually permeate is larger than the membrane thickness Δ*x*, and an equivalent pore radius could be calculated from Equation (4). The concept that lengths of tritium-labeled water [3] HHO diffusion and hydraulic permeation through tortuous capillary pores are the same improved the quality of the model. Sakai et al. provided the following correlation equation for estimating solute permeability [5,6]:(1)$$P_{m}=D_{O}f\left(q\right)S_{D}\frac{A_{K}}{\tau \Delta x}$$

In addition, τ is represented as
(2)$$\tau =\frac{H}{A_{K}}$$

Equation (4) is obtained by rearranging Equation (3)
(3)$$L_{P}=\frac{r_{P}^{2}H}{8\mu \tau^{2}\Delta x}$$
(4)$$r_{P}=\sqrt{\frac{8\mu \tau^{2}\Delta x}{H}\cdot L_{P}}$$
where *D*_O_ is the diffusion coefficient of the solute in water, *f*(*q*) is the friction coefficient of the pore walls and water with the solute in diffusion, *S*_D_ is the steric hindrance factor at the pore inlet in diffusion, *A*_K_ is the surface porosity of the water content, τ is the tortuosity, Δx is the membrane thickness, *H* is the water content, *L*_P_ is the pure water permeability, *r*_P_ is the pore radius, μ is the viscosity of pure water. The viscosity of pure water is 0.695 cP. The tortuosity of the membrane used was assumed to be 1.6 empirically [15]. Many of the membranes used for hemoconcentrator are asymmetric membranes, strictly speaking the tortuous pore diffusion model cannot be applied to asymmetric membranes that have a skin layer or another similar region that contributes largely to separation. 

### 4.3. Comparison of Pore Diameters Calculated by the Tortuous Pore Diffusion Model (TPD Model) and Measured by SPM

Table 4 shows the values for pore diameter before and after blood contact, which were calculated based on the TPD model, and the pore diameter before blood contact, which was measured using SPM.

The average pore diameter measured using a SPM before blood contact and that calculated by the TPD model almost matched, and both values were validated mutually. From this, the pore structure of the thick part of the membrane of the black box after blood contact could be estimated using the TPD model. Thus, the TPD model is significant because the pore structure of the membrane is associated with pure water permeability and can be inferred quantitatively.

The water content of the ultrafiltration membrane with high L_p_ and with low L_p_ was 79 ± 14 and 77 ± 18 *v/v*%, respectively (*n =* 10) [16,17,18,19]. The standard deviation was high (large variation), and no significant differences were found. A pending issue is the difficulty in accurately measuring the water content of hydrophobic membranes. We used the values, and the resulting pore diameters calculated through Equation (4) (where the TPD model values were 30.5 and 23.1 nm, respectively), under an assumed tortuosity of 1.6 [15]. 

The average pore diameters based on SPM were smaller than those determined with the tortuous pore diffusion model. The two polyethersulfone membranes described here were both asymmetrical and showed structures in which the pore diameter (structure) was minimum in the innermost surface facing the blood adsorption layer and outside surface [12]. By contrast, in the TPD model, the membrane structure is treated as homogenous [5,6], the pore shape is considered a perfect circle, and a pore diameter is considered consistent. Therefore, the pore diameter calculated using the TPD model (reflecting the micropore structure for the overall membrane thickness using the actually measured pure water permeability) is thought to be slightly larger than the surface pore diameters of inner lumen of the capillary membranes measured using SPM.

## 5. Conclusions

High-resolution 3D pore structures could be observed on the surface of inner lumen of the capillary membranes for hemoconcentration before blood contact. In addition, we observed, for the first time, that the protein adsorption layer covered the innermost surface pores after blood contact. The pore diameters of the thick part of the membrane that were calculated using the TPD model and those of the surface of inner lumen of the capillary membranes observed by SPM were mostly the same and could be validated reciprocally. Achievable pure water permeabilities showed no difference despite protein fouling on the pores, and relatively constant filtration rates make these capillary membranes suitable for clinical use. This approach using the TPD model and a SPM system, as shown in this study, is very useful for quantitatively analyzing the pore structure and properties of the membrane during fouling.

## Figures and Tables

**Figure 1 membranes-10-00315-f001:**
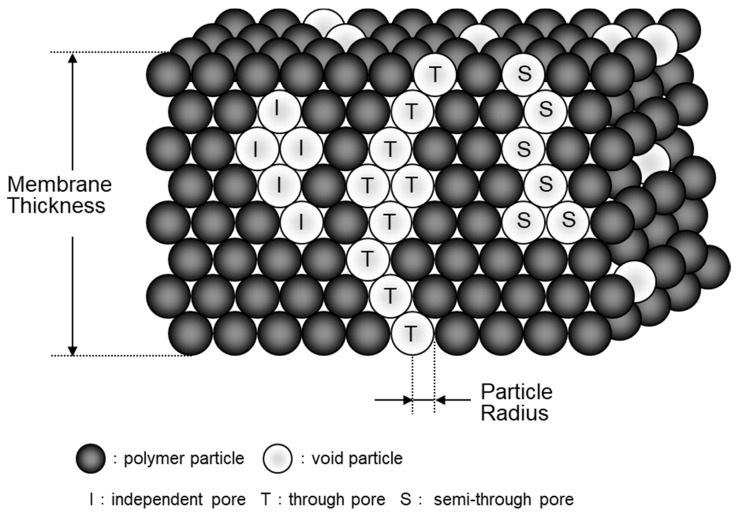
Schematic diagram of a three-dimensional laminated structure model [4].

**Figure 2 membranes-10-00315-f002:**
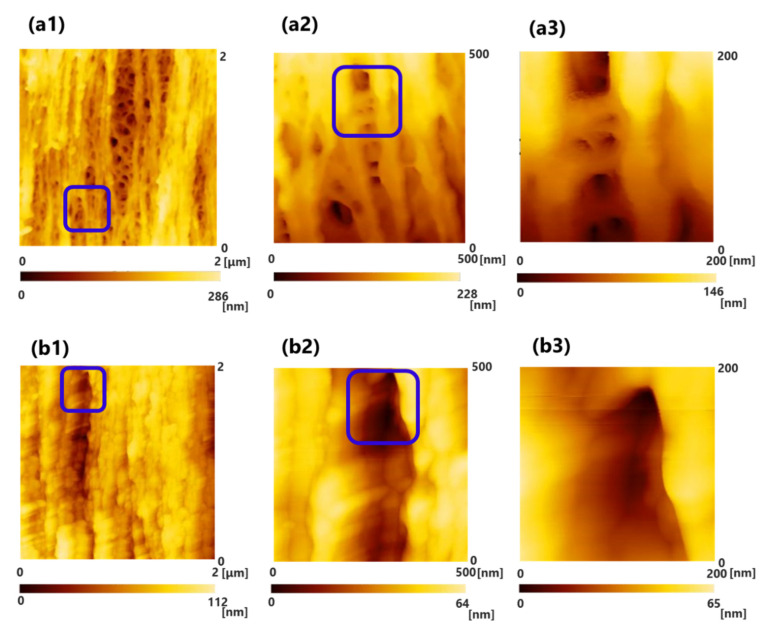
Scanning probe microscopy (SPM) images of the surface of inner lumen of the capillary membranes of PUREMA A (wet) (**a1**–**a3**) before blood contact, (**b1**–**d3**) after blood contact, (**b1**–**c3**) low load in blood contact, (**d1**–**d3**) high load in blood contact, respectively. (1) 2000 nm × 2000 nm; (2) 500 nm × 500 nm; and (3) 200 nm × 200 nm. The image of (3) shows magnified images of the pores enclosed by the blue square in (2), and the image of (2) shows magnified images of the pores enclosed by the blue square in (1). (**a1**): Clear three-dimensional (3D) pore structures and fiber structures made of polymer were observed before blood contact. (**d1**): Protein adsorption layer appeared to be thicker than those of (**b1**,**c1**).

**Figure 3 membranes-10-00315-f003:**
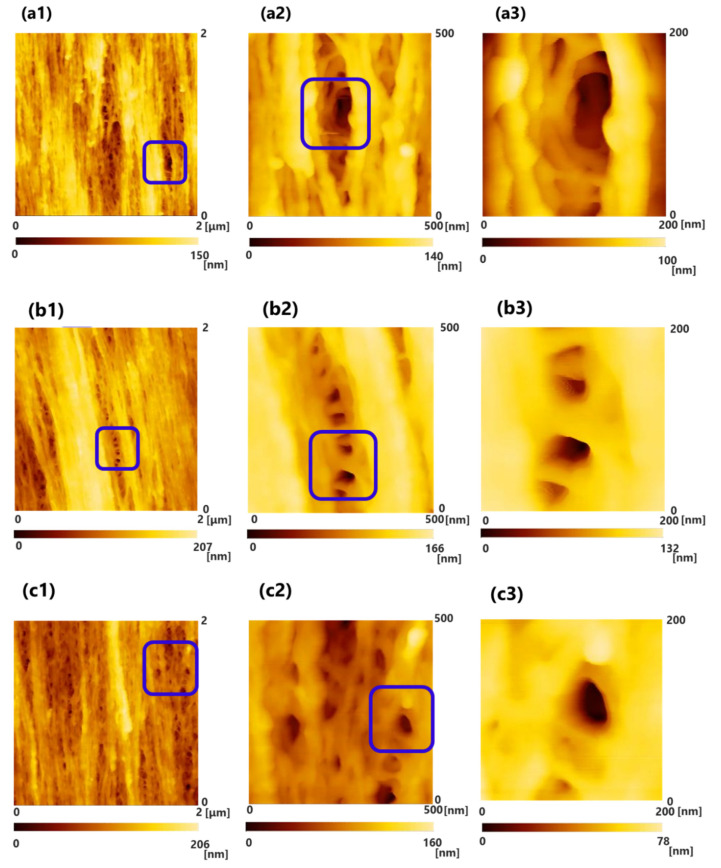
SPM images of the surface of inner lumen of the capillary membranes of PUREMA B (wet) (**a1**–**a3**) before blood contact, (**b1**–**d3**) after blood contact, (**b1**–**b3**) low load in blood contact, (**c1**–**c3**) high load in blood contact, respectively. (1) 2000 nm × 2000 nm; (2) 500 nm × 500 nm; and (3) 200 nm × 200 nm. The image of (3) shows magnified images of the pores enclosed by the blue square in (2), and the image of (2) shows magnified images of the pores enclosed by the blue square in (1), (**a1**–**a3**): cited from [20].

**Figure 4 membranes-10-00315-f004:**
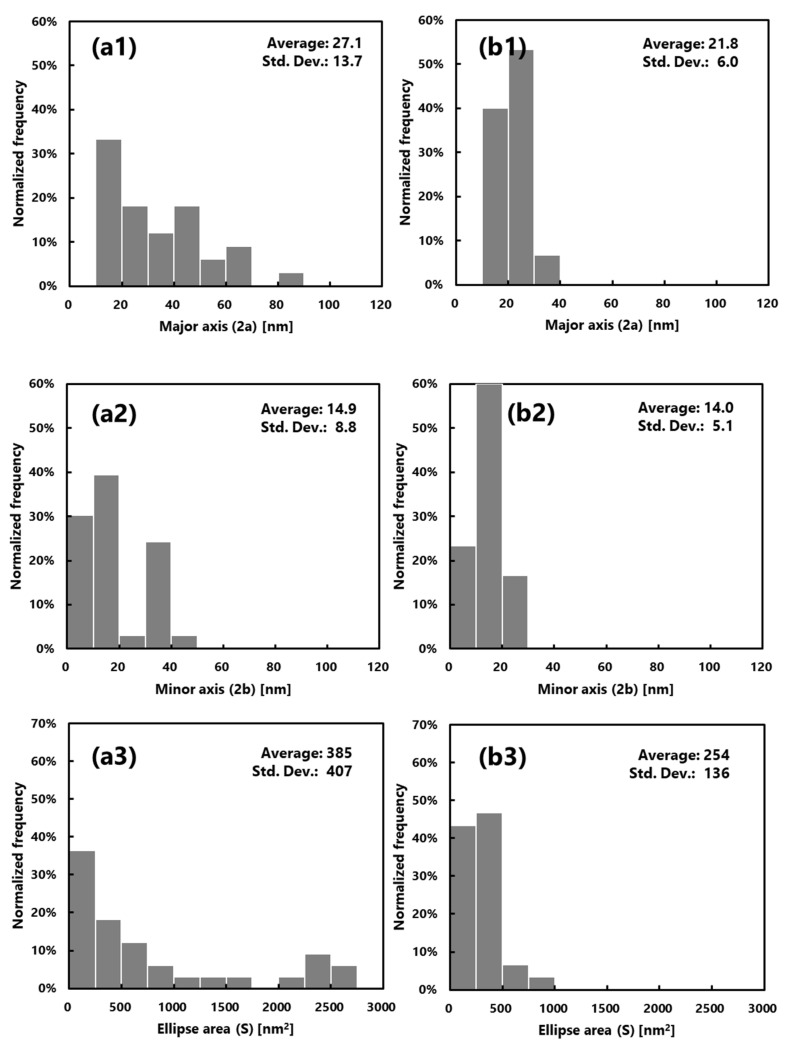
Distribution of the pore diameter before blood contact determined through SPM: (**a1**–**a3**) PUREMA A (wet) and (**b1**–**b3**) PUREMA B (wet). (1) Major axis, (2) minor axis, (3) pore area.

**Figure 5 membranes-10-00315-f005:**
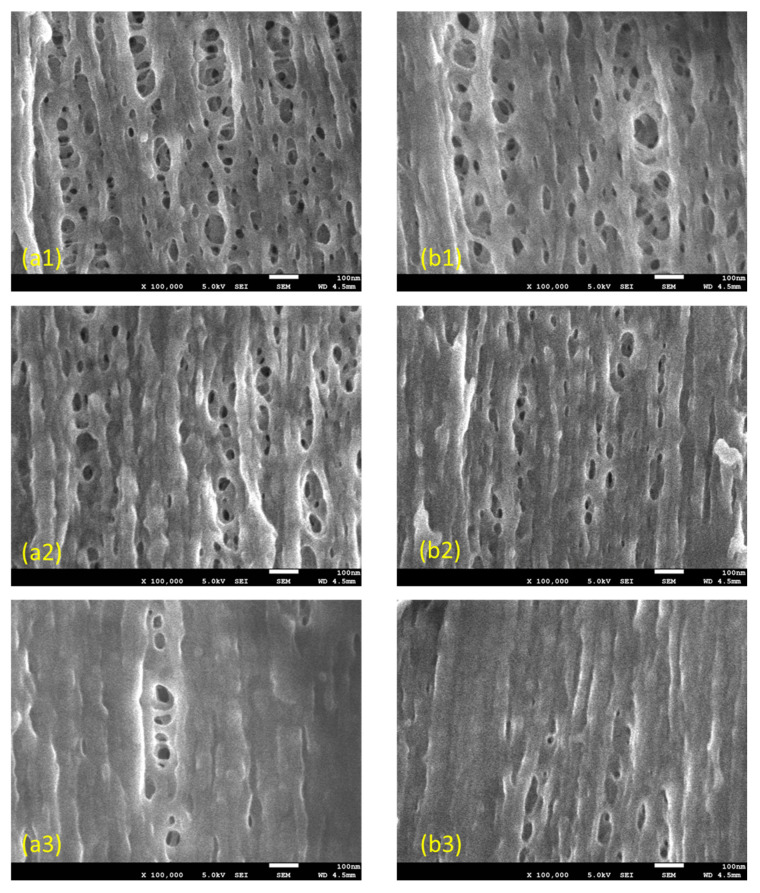
FE-SEM images of the inner lumen surface of the capillary membranes (**a1**–**a3**) PUREMA A (wet) and (**b1**–**b3**) PUREMA B (wet), (1) before blood contact, (2,3) after blood contact, (2) low load in blood contact, (3) high load in blood contact, respectively.

**Figure 6 membranes-10-00315-f006:**
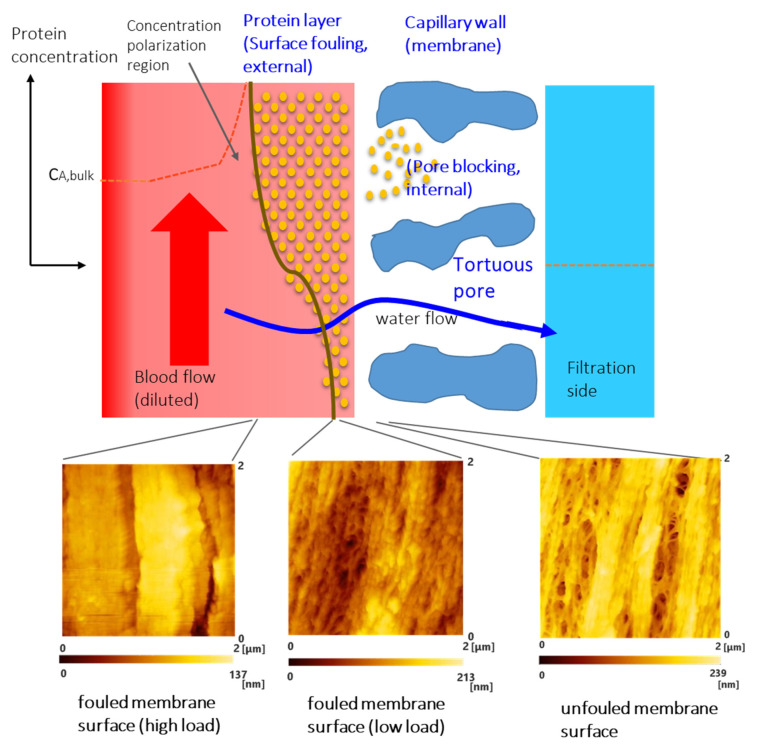
Schematic diagram of the fouling and pore blocking on the inner lumen surface of the capillary membrane (model).

**Table 1 membranes-10-00315-t001:** Specification of hollow fiber hemoconcentrator membranes tested.

Membrane Tested	Manufacturer	Pure Water Permeability [mL/hr/mmHg/m^2^]	Inner Diameter [μm]	Wall Thickness [μm]	Sterilization Condition	Sterilization Method
PUREMA A	3M Co., Ltd. (Aqua Stream^®^, JMS Co., Ltd. Japan)	207 ± 5	200	30	Dry	Ethylene Oxide Gas (EOG)
PUREMA B	3M Co., Ltd. (Aqua Stream^®^, JMS Co., Ltd. Japan)	115 ± 2	200	30	Dry	Ethylene Oxide Gas (EOG)

Membrane materials: polyethersulfone, hydrophilized by blending poly(vinylpyrrolidone) (PVP), glycerol free; membrane area of hemoconcentrator: 1.1 m^2^

**Table 2 membranes-10-00315-t002:** Observation conditions of field emission scanning electron microscope (FE-SEM).

Equipment	JSM-7610F (Jeol Ltd., Japan)
Observation mode	Secondary electron image
Accelerating voltage	1.5 kV
Emission current	47.2 µA
Working distance	4.5 mm
Magnifications	100,000

**Table 3 membranes-10-00315-t003:** Changes in pure water permeability before and after blood contact, sieving coefficient.

Sample	Pure Water Permeability L_p_ (mL/hr/mmHg/m^2^) ^(1)^			Sieving Coefficient of Albumin (-) ^(2)^
	Before Blood Contact	After Blood Contact (High Load)	After Blood Contact (Low Load)	
PUREMA A	207 ± 5	180 ± 4 *	194 ± 3 *	0.01 ± 0.02
PUREMA B	115 ± 2	114 ± 1 *	116 ± 1	0.02 ± 0.01

(1) *n =* 10, AVG. ± STD. Pure water permeability L_p_, Japan Society for Artificial Organs “Dialyzer Performance Evaluation Criteria” II. UFR measurement method, C method (STOP method, dead-end filtration); (2) *n =* 3, (-): dimensionless number, AVG. ± STD. * Paired t-test (*p* < 0.05, PUREMA A before blood contact versus after blood contact (high, low)), paired t-test (*p* < 0.05, PUREMA B before blood contact versus after blood contact (high)). Public criteria have not been established for the mass transfer properties of hemoconcentration membranes.

**Table 4 membranes-10-00315-t004:** Changes in the pore diameter before and after blood contact of the hemoconcentration membrane.

Sample	Pore Diameter Calculated from the Tortuous Pore Diffusion Model (nm)			Pore Diameter as Observed with SPM (Before Blood Contact) ^(1)^		
	Before Blood Contact	After Blood Contact (High Load)	After Blood Contact (Low Load)	Long Axis (nm)	Short Axis (nm)	Mean (nm)
PUREMA A	30.5	28.5	29.5	27.1 ± 13.7 *	14.9 ±8.8 *	24.6
PUREMA B	23.1	22.9	23.1	21.8 ± 6.0	14.0 ± 5.1	17.9

(1) *n =* 30, AVG. ± STD. * Paired t-test (*p* < 0.05, PUREMA A versus PUREMA B).

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
