# Peer review of "Validity of Three-Dimensional Tortuous Pore Structure and Fouling of Hemoconcentration Capillary Membrane Using the Tortuous Pore Diffusion Model and Scanning Probe Microscopy"

_membranes, 2020, doi:10.3390/membranes10110315_

Round 1

Reviewer 1 Report

This is a revised submission. My main criticism before of the paper was "In general I found the article somewhat difficult to read and I also found the apparent lack of protein fouling on the low Lp capillaries hard to understand. "

The paper is still difficult to read, but this time I realized it was partly due to the author's frequent use of the term thickness of the adsorbed protein layer, without any explanation of how they determined or assessed thickness. 

But my main question, "Why would there be protein fouling on high Lp capillaries but not low Lp capillaries? The chemical composition of the capillaries is the same. The authors never address this apparent difference, but they should. “went basically unanswered by the authors, who seemed content to add only the statement that the answer to my question was unknown.  I found that pretty unsatisfactory.

They chose to add another figure using SEM imaging of the surfaces that they said confirmed the original results obtained with SPM.  However, they also commented that they detected degradation of the surfaces by heating in the SEM imaging studies, so I do not think these doubtful images should be included.

Reviewer 2 Report

The manuscript presented by the authors describes investigations regarding hemoconcentration capillary membranes. The general concept of the work is convincing, although the investigated problem is highly specialized. It would be interesting to see if the model and technique applied here could be used for other fouling problems as well, making the overall idea interesting to a bigger audience.

A few minor errors were found by the reviewer, which should be corrected before publication:

  • Line 32 „albumin SC“ abbreviation should be explained
  • Line 55 “model. [4].” remove additional period
  • Line 201 “QCM-D” abbreviation should be explained
  • Line 223 ”The pore and area distributions were also shown to appear large and wide (the pore and area distributions were sharp).” This sentence is confusing, please rephrase.
  • Line 285 “[3].1 h” space missing
  • Line 315-317 “(1951)” is mentioned twice, one can be removed
  • Line 345 “table 3” should be “table 4”

Reviewer 3 Report

The manuscript by Fukuda et al describes the three-dimensional tortous pore structure and fouling of hemoconcentration capillary membrane. Additionally the tortuous diffusion model Scanning probe microscopy was used and compared.

Several Aspects of the paper are interesting but further discussion is needed before it should be published. Therefore, I suggest major revision.

  • Introduction: Figure 1 should be described more in the text.
  • The Introduction with respect to hemoconcentration membranes is a bit low. In order to satisfy the broad readership of “Membranes” the first paragraph should include general aspect about these type of membranes before going into details. In addition topics like where fouling causes problems special for this membranes should be mentioned
  • Part 2: The format of Table 1 does not look good

In general: mmHg as unit should be changed to the international system of units in the future, not mandatory here, but would be better

  • Protein Fouling: It would be interesting to determine the fouling rate by UV measurement Therefore the membranes could be soaked in protein solution for a defined time and the protein concentration of the solution should be detect befor and after adsorption.
  • Is the fouling in this case reversible? How are the pemeances after the membranes are washed?
  • Permeances: You could add a graph shown the permeances over a certain time
  • 3: Comparison of pore diameters calculated by the ….and measured by SPM (please correct)

Best wishes

Round 2

Reviewer 3 Report

Dear Authors,

please explain in details how permeabilities were measured, if you are not able to measure the permeability change over time (this,I  would recommend by the way). Please add how it was measured , how long, how long were the membrane exposed to the media before measurements. The temperature plays also a role.

Best regards

Round 3

Reviewer 1 Report

This a second revision of a manuscript I reviewed. It is now considerably improved in response to my last set of commeny and I have no further comments.

This manuscript is a resubmission of an earlier submission. The following is a list of the peer review reports and author responses from that submission.

Round 1

Reviewer 1 Report

In this work, the authors used scanning probe microscope method and tortuous pore diffusion model to study three-dimensional pore structures and protein fouling of the capillary membranes before and after blood contact. Results showed that the pore diameters calculated by TPD and SPM were mostly similar and could be validated reciprocally. They found that the pure water permeability of capillary membranes before and after blood contact showed no difference despite protein fouling, because fouling occurred on the membrane surface and internal pores were almost not blocked. Overall, the work is lack of originality, and I think the research contents are not suitable for Membranes. In addition, there are some problems in the manuscript.

  1. In section 3.1, the authors used the SPM line analysis to study the protein adsorption layer of PUREMA A and PUREMA B membranes under high and low load in blood contact. However, SPM results are greatly affected by the selection of measurement area, and cannot be used for quantitative analysis of protein adsorption. So, the authors should try other experimental instruments to test the protein adsorption on the surface of capillary membranes, such as QCM-D measurements.
  2. In section 4.3, the pore diameters of capillary membranes measured by TPD model and SPM were compared. The authors found that the average pore diameters based on SPM were smaller than those determined with TPD model. But this conclusion is already published in their previous work (Advanced Biomedical Engineering, 8:145-152).
  3. It is suggested that the authors should perform more experiments about hydrophilic materials to modify the capillary membranes, so as to improve the innovation of the work.

Reviewer 2 Report

The authors presented scanning probe microscopy (SPM) images of the inner surfaces of fibers from hemoconcentration devices made with capillaries, both before and after exposure to bovine blood. Two device types were studied, with approximately two-fold difference in water permeability (Lp) (207 vs 115), and each was exposed to two blood flow at rates (500 or 100 ml/min). Water permeability and sieving coefficient of albumin were also measured after blood contact. The main conclusions were that pores observed with SPM before blood contact on the high Lp capillaries tended to disappear after blood contact, especially at the higher blood flow rate, while the pores in the low Lp capillaries were still readily visible after blood contact.  In addition, there was little change in Lp or calculated pore diameter to any of the devices after blood contact.

In general I found the article somewhat difficult to read and I also found the apparent lack of protein fouling on the low Lp capillaries hard to understand. I give more detailed comments below.

  1. Why would there be protein fouling on high Lp capillaries but not low Lp capillaries? The chemical composition of the capillaries is the same. The authors never address this apparent difference, but they should.
  2. The authors conclude there must be no protein fouling on the internal pores of the capillaries because the permeability did not change after exposure to blood.  Why would there be no protein adsorption to the internal pores??
  3. There are numerous places in the text that are hard to understand or inappropriate, here are some examples:

Abstract :  “On the other hand, pore structures after blood contact could not be observed due to protein fouling.” That statement only applies to some of the results.

Abstract:  “This is due to the mechanism that protein fouling occurs on the membrane surface, while there is little internal pore blocking.” No evidence was provided on protein fouling in the internal pores.

Intro: “These concepts have led to the development of membrane science, the development of blood purification, and the creation of a major medical device industry that involves polymer chemistry and chemical engineering. Membrane science is significant in terms of its contribution to engineering and blood purification [7]. The development of the medical device industry and membrane science is a two-way process.” These sentences are so general and irrelevant that they should be deleted.

Intro: “As an example, for lithium-ion secondary batteries, Yoshino (Akira Yoshino, 2019 Nobel laureate in chemistry, honorary fellow at Japanese chemical producer Asahi Kasei, the basic foundation of the lithium-ion battery was established.) et al. are studying the relation between the 3D pore structures of flat sheet membranes (separator) and lithium-ion migration [9-10].” These sentences are garbled and also of little relevance.

Methods: The anticoagulant used should be stated.

Results: The state the pores are not obscured by adsorbed protein on low Lp  capillaries. They then state: “So it has been assumed that the thickness of the protein adsorption layer of the ultrafiltration 182 membrane with low Lp was thicker.” These seem to be contradictory.

Results: They refer to blood cell adhesion as one of the steps in blood interaction. Why did they not observe adherent blood cells on the capillaries after blood exposure?

Figure legend 2, last part: “Table  1. appeared to be thicker than those of (b1, c1).” I have no idea what this is supposed to mean.

Reviewer 3 Report

Fukuda et al have investigated the characteristics of pores in homoconcentration membrane under in the presence of blood samples using mathematical and experimental approaches.

The topic is of great importance, data is presented statistically , methods is explained in a transparent way and in some cases the authors admit the shortcomings of their approaches.

My minor comments:

  1. Short axis and long axis are not well-defined, please explain it in detail.
  2. manuscripts needs to be edited carefully for the language.
  3. these sentences were so obscure and out of context.

 **we were interested in pore structures before and after blood 42 contact due to protein fouling [3].

**particles (Fig.1) refer to the model 45 according to Hiyoshi’s publication [4].

**Yoshino (Akira Yoshino, 2019 Nobel laureate 61 in chemistry, honorary fellow at Japanese chemical producer Asahi Kasei, the basic foundation of the 62 lithium-ion battery was established.) et al. are studying (lines 60-63)

**determined by another approach and a model of 73 the pore shape predicted…

  1. Tables need to be re-formatted.